# Nitrite Enhances MC-LR-Induced Changes on Splenic Oxidation Resistance and Innate Immunity in Male Zebrafish

**DOI:** 10.3390/toxins10120512

**Published:** 2018-12-03

**Authors:** Wang Lin, Honghui Guo, Lingkai Wang, Dandan Zhang, Xueyang Wu, Li Li, Dapeng Li, Rong Tang

**Affiliations:** 1College of Fisheries, Huazhong Agricultural University, Wuhan 430070, China; linwang@webmail.hzau.edu.cn (W.L.); honghuiguo@webmail.hzau.edu.cn (H.G.); wanglingkai@webmail.hzau.edu.cn (L.W.); dan19950824@webmail.hzau.edu.cn (D.Z.); wuxueyang@webmail.hzau.edu.cn (X.W.); ldp@mail.hzau.edu.cn (D.L.); tangrong@mail.hzau.edu.cn (R.T.); 2Hubei Provincial Engineering Laboratory for Pond Aquaculture, Wuhan 430070, China; 3National Demonstration Center for Experimental Aquaculture Education, Huazhong Agricultural University, Wuhan 430070, China

**Keywords:** Microcystin-LR, Nitrite, oxidative stress, immune function, joint toxicity

## Abstract

Hazardous contaminants, such as nitrite and microcystin-leucine arginine (MC-LR), are released into water bodies during cyanobacterial blooms and may adversely influence the normal physiological function of hydrobiontes. The combined effects of nitrite and MC-LR on the antioxidant defense and innate immunity were evaluated through an orthogonal experimental design (nitrite: 0, 29, 290 μM; MC-LR: 0, 3, 30 nM). Remarkable increases in malondialdehyde (MDA) levels have suggested that nitrite and/or MC-LR exposures induce oxidative stress in fish spleen, which were indirectly confirmed by significant downregulations of total antioxidant capacity (T-AOC), glutathione (GSH) contents, as well as transcriptional levels of antioxidant enzyme genes *cat1*, *sod1* and *gpx1a*. Simultaneously, nitrite and MC-LR significantly decreased serum complement C3 levels as well as the transcriptional levels of splenic *c3b*, *lyz*, *il1β*, *ifnγ* and *tnfα*, and indicated that they could jointly impact the innate immunity of fish. The severity and extent of splenic lesions were aggravated by increased concentration of nitrite or MC-LR and became more serious in combined groups. The damages of mitochondria and pseudopodia in splenic macrophages suggest that oxidative stress exerted by nitrite and MC-LR aimed at the membrane structure of immune cells and ultimately disrupted immune function. Our results clearly demonstrate that nitrite and MC-LR exert synergistic suppressive effects on fish innate immunity via interfering antioxidant responses, and their joint toxicity should not be underestimated in eutrophic lakes.

## 1. Introduction

During the last decade, the issue of harmful cyanobacteria blooms has become a worldwide concern since such blooms may cause oxygen depletion of water body and release cyanotoxins [1,2,3]. Microcystins are a family of cyclic heptapeptide toxins released by cyanobacteria [4]. Among approximately 100 microcystin isoforms, microcystin-leucine arginine (MC-LR) is considered to be the most toxic and widely distributed in freshwater bodies [5,6]. The toxins are environmentally stable and can persist in aqueous for days due to the cyclic structure [7,8]. For instance, the MC content in lake water varied from traces up to 35.8 μg/L (36.02 nM) in Lake Taihu of China after the collapse of cyanobacterial blooms [9], posing an enormous health hazard to aquatic animals, terrestrial livestock and even humans [10,11,12]. Moreover, the inhibition of serine/threonine protein phosphatase 1 and 2A is considered to be a classic toxic mechanism of MCs [13].

In particular, more attention should be paid to post-bloom toxicity resulted by the subsequent physical and bacterial degradation. Elevated ambient nitrite is a by-product of bloom events, which derives from the imbalance between bacterial nitrification and denitrification under anoxic conditions [14]. A nitrite level of 18 mg/L (260.87 μM) was reached in water bodies during the algae decaying period [15]. Previous studies demonstrated that nitrite exposure reduced growth, disturbed osmoregulatory function and caused hypoxia stress by inhibiting blood oxygen-carrying capacity in teleost fish [16,17,18,19]. The hypoxic conditions caused by nitrite could also damage various organs such as gill, liver and kidney [20]. To date, the available data have suggested that oxidative stress is one mechanism of the underlying nitrite toxicity [21]. However, most of the investigations on the toxicity of nitrite have been conducted as acute toxicity studies with high dosage use.

Aquatic organisms try to ameliorate oxidative damage by utilizing the antioxidant enzymes such as catalase (CAT), superoxide dismutase (SOD), glutathione peroxidase (GPx) and non-enzymatic antioxidants such as glutathione (GSH) [22,23,24]. Lipid peroxidation (LPO) usually occurs when reactive oxygen species (ROS) production exceeds the capacity for resistance of antioxidant systems of organisms. Previous research has shown that MC-LR has the potential to induce the production of LPO and subsequently influence the activities of CAT, SOD and GPx in different organs of fish including the liver, kidney and even ovary [25,26,27]. Acute nitrite stress has also been reported to cause oxidative damage and reduce antioxidant enzymes activities in fish liver [28]. Nevertheless, aquatic animals are most likely to be exposed to MC-LR combined with other water pollutants, such as nitrite, rather than to individual compounds in blooms area.

The incidence of oxidative stress resulting from exposure to environmental pollutants is linked to the modulation of immune function [29,30]. In general, fish largely rely on the innate immune defense, which involves both immune cells in the spleen and immune molecules dissolved in body fluids [31]. Recent studies revealed that MC-LR can accumulated in the fish spleen, induce pathological damages and suppress immune function in medaka fish (*Oryzias latipes*) and crucian carp (*Carassius carassius*) [32,33]. A few previous studies reported that high levels of nitrite could cause immunity dysfunction by impairing the immune-related enzymes and genes in fish [34,35]. Chand and Sahoo [36] documented that acute nitrite exposure reduced the immune function of *Macrobrachium malcolmsonii* to resist *Aeromonas hydrophilia* infection. Even though studies showed that nitrite or MC-LR alone could disrupt the immune function of aquatic animals, information about the joint toxicity and intoxication mechanism of nitrite and MC-LR on fish defense system has not been available.

Based on the above, our present study used zebrafish as a test organism and attempted to elucidate the joint effects of environmental levels of nitrite and MC-LR on the antioxidant and innate immune system as well as the detailed molecular mechanism. Our study provides new viewpoints to the combined toxicity of MC-LR and coexisting compounds on fish immune system during the period of cyanobacterial bloom.

## 2. Results

### 2.1. Nitrite and MC-LR and Concentrations in Water Samples

The measured concentrations of nitrite and MC-LR are shown in Appendix A. The results showed the measured concentrations of nitrite at 29 μM and 290 μM varied from 26.8 μM to 33.3 μM and from 258.1 μM to 332.1 μM, respectively. The real concentrations of MC-LR at 3 nM and 30 nM varied from 2.8 nM to 3.4 nM and from 26.8 nM to 33.5 nM, respectively. The maximum deviation between measured concentrations of nitrite or MC-LR and the corresponding nominal concentrations in each treatment group was less than 20%, which proved that the concentrations of these two contaminants within the experimental tanks were relatively stable during the whole exposure period.

### 2.2. Spleen Index

As shown in Appendix A, and Table 1, the spleen index increased significantly in both nitrite-treated groups and MC-LR-treated groups. Compared with the control group, a 48.2% maximal increase on the spleen index was observed in the co-exposure group of 290 μM nitrite and 30 nM MC-LR. Nevertheless, no statistically significant interaction was observed between nitrite and MC-LR on the spleen index.

### 2.3. The Levels of LPO, T-AOC and GSH in the Spleen

Splenic LPO levels (measured as malondialdehyde (MDA)) for fish treated by single concentration of nitrite or MC-LR increased significantly with increasing exposure concentrations (Table 1, Figure 1A). Similarly, the combinations of nitrite and MC-LR had significant influences on MDA, and the largest induction was 170% in the co-exposure group of 290 μM nitrite and 30 nM MC-LR. A two-way ANOVA demonstrated a marked interaction between nitrite and MC-LR on MDA. 

Single exposure to nitrite or MC-LR decreased total antioxidant capacity (T-AOC) significantly in the spleen, and T-AOC levels decreased by 29% and 30% in 290 μM in the nitrite-treated group and 30 nM in the MC-LR-treated group, respectively. The maximal decrease was 59% in the co-exposure group of 290 μM nitrite and 30 nM MC-LR (Table 1, Figure 1B). In contrast, however, there was no significant interaction of nitrite and MC-LR on T-AOC.

GSH content showed either an increasing or decreasing trend with increasing exposure concentrations of nitrite and MC-LR. However, no significant differences in GSH content were detected among the other exposure groups, except in the control group. Compared with the control, the contents of splenic GSH in zebrafish exposed to nitrite or MC-LR exhibited marked decreases, of which the largest reduction was up to 58% in co-exposure groups. Moreover, there was a significant interaction between nitrite and MC-LR on GSH content (Table 1, Figure 1C).

### 2.4. Serum Complement C3 Content

Significant reductions in serum complement C3 were observed in zebrafish exposed to single concentrations of nitrite. Additionally, single MC-LR caused a significant decrease in complement C3 content, which decreased by 65% in the 30 nM MC-LR group. The greatest decrease in C3 content was 78% in the highest co-exposure group, and a significant interaction were found between nitrite and MC-LR on complement C3 content (Table 1, Figure 1D).

### 2.5. mRNA Expression Profiles of Antioxidant and Innate Immune-Related Genes in the Spleen

To evaluate the antioxidant and nonspecific immune responses to MC-LR in combination with nitrite, related gene transcriptions were determined in the spleen (Table 2, Figure 2). A single concentration of MC-LR had a significant inhibition on the transcriptional levels of *cat1*, *sod1* and *gpx1a*, whereas nitrite only significantly decreased *gpx1a* mRNA. A significant interaction was detected between nitrite and MC-LR on the transcription level of *gpx1a*. For innate immune-related genes, including *c3b, lyz, il1β*, *ifnγ* and *tnfα*, transcriptional levels were downregulated significantly by high concentrations of nitrite and MC-LR, although they showed a mild increase after the low-concentration exposure to single nitrite or MC-LR. Moreover, there were significant interactions between nitrite and MC-LR on these innate immune-related genes.

### 2.6. Splenic Pathological Observation

In the control group, the spleen of zebrafish showed a normal appearance with abundant erythrocytes and leukocytes as well as some macrophages (Figure 3A and Figure 4A). In MC-LR-treated groups, an elevation in the number of melano-macrophage centers was noticed with increasing exposure concentration (Figure 3B). Using transmission electron microscopy, we observed that melano-macrophage centers consisted of macrophages with different sizes of swallowed masses, and these macrophage pseudopodia became degenerative with the increase of MC-LR concentration (Figure 4B,C). In nitrite-treated groups, the splenic tissue was filled with a great deal of erythrocyte, indicating a symptom of blood stasis (Figure 3C). Similar to the results from light microscope observation, ultrastructural observation showed plenty of erythrocytes and hydropic mitochondria in macrophages (Figure 4D,G). Moreover, combined concentrations of nitrite and MC-LR had similar but more severe influences on splenic structure, which were characterized with erythrocyte stasis, pseudopodia degeneration and mitochondria vacuolization of melano-macrophages (Figure 3D and Figure 4E,F,H,I).

### 2.7. Correlation Analysis

As shown in Appendix A, innate immune parameters *il1β*, *ifnγ*, *tnfα* and C3 were negatively correlated with MDA contents, and positively correlated with T-AOC levels. The mRNA levels of innate immune-related genes including *il1β*, *ifnγ*, *tnfα*, *c3b* and *lyz* were positively correlated with the expression of *sod1* and *gpx1a*. Moreover, *tnfα*, *c3b* and *lyz* showed positive correlation to *cat1.*

## 3. Discussion

Aquatic organisms are rarely exposed to individual chemicals in isolation but rather to a complex mixture of chemicals in natural waters [37]. With the worldwide occurrence of cyanobacterial blooms, however, there is no information about the joint effects of MC-LR and other water pollutants like nitrite on fish immune system. The immune system is very important for aquatic organisms, since its impairment ultimately leads to increased host susceptibility to infectious diseases [38]. The present study was the first to provide well-founded evidence for the effects of combined factors (nitrite & MC-LR) on splenic oxidant-antioxidant status and innate immune responses in male zebrafish.

### 3.1. Effects of Nitrite and MC-LR on Splenic Oxidant-Antioxidant Status

MDA, the final product of membrane lipid peroxidation, is a sensitive diagnostic index of oxidative damage [39,40]. In the present study, a dose-dependent increase of MDA in the spleen was found in the groups exposed to single factor nitrite or MC-LR, indicating that both nitrite and MC-LR promoted the production of ROS and induced oxidative stress. A further synergistic effect of nitrite and MC-LR was detected on splenic MDA and the maximum induction of splenic MDA was up to 170% in co-exposure group, which implies that MC-LR could induce more severe oxidative damages under nitrite condition. Detailed evidence has showed that nitrite and MC-LR could induce an increase of MDA level in fish liver [26,41], however, data are limited about the effects of nitrite and MC-LR on splenic MDA. As we know, the spleen is one of the important immune organs in fish. The oxidant-antioxidant balance is essential for splenic function, as it maintains the integrity and functionality of membrane lipids in immune cells [29]. Hence, oxidative stress caused by nitrite and MC-LR may play an important role in the toxicity of the two pollutants in fish spleen.

Consistent with the finding of splenic MDA, T-AOC levels in the spleen were significantly decreased in the groups exposed to single nitrite or MC-LR, suggesting that the fish’s antioxidant defense system were impaired by nitrite and MC-LR. Similar results were found in the liver of *Jenynsia multidentata* exposed to extract MCs of 5 and 100 mg/L (equivalent to 5.03 and 100.50 μM purified MC-LR) for 24 h [42]. A remarkable decrease in T-AOC level was detected in the liver of *Carassius auratus* exposed to sodium nitrite for 96 h, which indicates that acute nitrite exposure severely impacted hepatic antioxidant system [43]. Thus, our results along with previous studies revealed that the antioxidant capacity of fish spleen was greatly influenced by prolonged exposure of nitrite and MC-LR. Compared with the single-factor exposure groups, the decrease of splenic T-AOC was up to 59% in the co-exposure groups, which confirmed synergistic interactions of nitrite and MC-LR on T-AOC. Previous studies reported that hepatic antioxidant enzymes (CAT, SOD and GPx) in fish could help to eliminate excessive ROS induced by nitrite and MC-LR, whereas these enzyme activities and transcriptional levels were significantly inhibited after exposure to high-concentration nitrite or MC-LR [26,44,45]. In fish spleen, the antioxidant defense system plays an essential role in alleviating oxidative stress [46]. Along with decreased T-AOC contents, transcription levels of *cat1*, *sod1* and *gpx1a* in the spleen showed concurrent suppression after single and combined treatments with nitrite and MC-LR, which indirectly proved the overproduction of ROS. Furthermore, splenic GSH content, an important non-enzyme antioxidant, showed a significant reduction after exposure to nitrite, MC-LR or their mixture in the present study. Additionally, an interaction effect on splenic GSH was detected between nitrite and MC-LR. In fact, the maintenance of a necessary amount of GSH is vital for fish to protect themselves from harmful pollutants and consequent ROS [47,48]. Li et al. [47] demonstrated that GSH levels decreased significantly along with ROS formation in the hepatocytes of common carp (*Cyprinus carpio* L.) after exposure to MC-LR, which suggests that GSH might play key roles in alleviating oxidative damage and the detoxification process of MC-LR. Jia et al. [49] found a considerable reduction of GSH in the gill when juvenile turbot (*Scophthalmus maximus*) were exposed to 0.8 mM nitrite for above 24 h. Therefore, the depletion of splenic GSH caused by nitrite and MC-LR in our study implied that the detoxification capability of fish was greatly inhibited when fish encountered the concurrence of these two toxins.

### 3.2. Effects of Nitrite and MC-LR on Innate Immune Responses

The lysozyme and complement are two crucial innate immune molecules in fish and often used as terminal indexes of the innate immunity [50,51]. The main function of lysozyme is to destroy and eliminate the invasion of xenobiotics by forming a hydrolytic enzyme system [52,53]. Fish complement can lyse foreign cells and opsonize foreign organisms for destruction by phagocytes, meanwhile the third complement component (C3) plays a central role in the complement cascade [54]. Pure MC-LR caused a marked decrease in the activity and transcript level of lysozyme in the hepatopancreas of white shrimp (*Litopenaeus vannamei*), which made them more susceptible to pathogens [55]. Similarly, serum lysozyme activity significantly decreased in *Labeo rohita* juveniles exposed to 2 mg/L (29 μM) nitrite [56]. To date, very few studies have been focused on the effects of nitrite or MC-LR on complement C3. In the present study, exposure to nitrite or MC-LR alone significantly decreased serum C3 as well as mRNA levels of splenic *c3b* and *lyz*, indicating the immune toxicity induced by nitrite and MC-LR. Furthermore, the co-exposure of nitrite and MC-LR aggravated the downregulation of C3 and lysozyme in protein and/or gene levels and suggested a synergistic immune suppression of nitrite and MC-LR. 

The present findings also showed that transcriptional levels of cytokine genes in the spleen, including *il1β*, *tnfα* and *ifnγ*, were induced by low concentration of single nitrite or MC-LR, however, they were inhibited by their respective high concentration. The main cytokines, such as interleukin-1β (IL-1β), tumor necrosis factor-α (TNF-α) and interferon-γ (IFN-γ) secreted by immune cells play a pivotal role in modulating the inflammatory response of fish to pathogenic germs or toxicants [57,58]. Rymuszka and Adaszek [59] reported that pure MC-LR and MCs extract significantly upregulated the expression of pro-inflammatory cytokines (*il1β* and *tnfα*) in blood phagocytes of common carp (*Cyprinus carpio* L.) and indicated the activation of the inflammatory response. Similarly, the mRNA expression of IL-1β and TNF-α genes increased significantly in the gills of turbot exposed to 0.8 mM nitrite for 96 h [35]. An opposite downregulation in mRNA levels of these cytokines was reported in the spleen of grass carp (*Ctenopharyngodon idella*) intraperitoneally injected with 50 μg MC-LR·kg^−1^ [60]. The discrepancy on different responses of cytokines in fish exposed to environmental chemicals might be dependent on the exposure dose and exposure manner [35,60,61]. Thus, the dualistic changes of cytokine mRNA levels might suggest concentration-dependent immunomodulatory effects of single nitrite or MC-LR. The innate immune responses of fish are stimulated when the stressor is short-term, whereas the immune capacity shows suppressive effects when the stressor is chronic and long-term [62]. Jin et al. [61] reported that the over-expression of *il1β* and *tnfa* mRNAs induced by permethrin and estradiol likely resulted in decreased anti-infection effects of immune-related cells in zebrafish embryos. From this point, the long-term inflammatory responses after exposure to low concentrations of nitrite or MC-LR might eventually influence the ability to fight infections and toxicants. Our co-exposure results clearly demonstrated that nitrite and MC-LR had interactive repressive effects on these cytokines at the mRNA level.

The immune cell functions are potently regulated by the equilibrium of oxidation and antioxidation, hence, their antioxidant status helps to preserve the normal functions against homeostatic disturbances caused by oxidative stress [63,64]. According to the correlation analysis, MDA content exhibited negative correlation with innate immune parameters, implying that the compromised innate immunity should be attributed to oxidative stress after exposure to nitrite and MC-LR. Previous studies also demonstrated that increased endogenous ROS in immune cells impacted cell-mediated immune function, and subsequently impaired immunocompetence [29,65]. Given all that, our present results revealed that nitrite and MC-LR might, either separately or in combination disrupt innate immune function by interfering the antioxidant system.

### 3.3. Effects of Nitrite and MC-LR on Splenic Pathological Changes

Fish immunity is correlated with the normal structure and function of immune organs [66], thus, the damage of these organs eventually results in the decrease of the resistance to toxic chemicals [67]. In the present study, the significant increase in spleen indexes implies that the single and combined exposure of nitrite and MC-LR caused spleen damage. The spleen is the largest immune organ in fish and contains a lot of lymphocytes and macrophages [68]. Macrophages play an important role in the innate immune system, since they phagocytose heterogeneous particles like microorganism, macromolecules, injured or apoptosis tissues [69]. Palíková et al. [70] revealed that the phagocytic ability of blood leucocytes was highly suppressed in silver carp (*Hypophthalmichthys molitrix* Val.) after oral administration of 1200 μg·kg^−1^ cyanotoxin. In the present study, MC-LR alone caused the pseudopodia degeneration and mitochondria vacuolization in splenic melano-macrophages, which revealed an immunosuppressive toxicity of MC-LR. The immune cells are particularly sensitive to oxidative stress because of the high percent of polyunsaturated fatty acids in their plasma membranes [29]. Moreover, mitochondria are the main places of endogenous ROS production, which are in turn vulnerable to ROS attacks [71]. Thus, the membrane damage of macrophages (mitochondria and pseudopodia) in MC-LR exposure groups further confirmed that MC-LR induced excessive ROS. As for the nitrite exposure, pathological changes were mainly characterized with erythrocyte stasis and hydropic mitochondria of macrophages. It was reported that the main toxicity of nitrite on aquatic animals is due to the conversion of oxygen-carrying pigments to forms that are incapable of carrying oxygen, and thus, causes hypoxia stress, and subsequently, elicit oxidative damage [72,73]. Thus, it was not surprising to observe hypoxic-induced polycythemia and stasis as well as hydropic mitochondria of the macrophages after exposure to nitrite. Furthermore, similar but more serious damage in the co-exposure groups proved direct evidence that nitrite and MC-LR caused synergistic oxidant-mediated effects.

## 4. Conclusions

The present study revealed that long-term single and combined exposure of nitrite and MC-LR (greater than or equal to 29 μM and 30 nm, respectively) could cause pathological damages, induce oxidative stress and inhibit innate immunity in the spleen of male zebrafish, indicating that fish might encounter severe immunotoxic effects during the degradation of cyanobacterial blooms. The study indicated that environmentally related concentrations of nitrite and MC-LR could jointly impair the antioxidant capacity and innate immunity of fish. The underlying mechanism of joint immunotoxicity of these two hazardous materials might result from the disequilibrium of oxidant-antioxidant homeostasis.

## 5. Materials and Methods

### 5.1. Chemicals

The desired concentration of nitrite was obtained by adding a stock solution of sodium nitrite (NaNO_2_, Shanghai, China). MC-LR (Purity ≥95% confirmed by HPLC) was purchased from Express Technology (TaiWan, China) and dissolved in deionized water. All the reagents utilized in this experiment were of analytical grade.

### 5.2. Zebrafish Maintenance, Exposure and Sampling

Wild-type male zebrafish (3-month-old) were obtained from the China Zebrafish Resource Center (CZRC), Chinese Academy of Sciences. During the two-week acclimation period, water temperature was maintained at 28 ± 0.5 °C and the photoperiod was adjusted to a 14 h light: 10 h dark regimen. Fish were fed with newly hatched *Artemia* three times daily.

The experiment was conducted in a three-level full factorial design and exposure concentrations (nitrite: 0, 29, 290 μM; MC-LR: 0, 3, 30 nM) were set in consideration of the environmental relevance [74,75]. After the acclimation, zebrafish were divided into nine treatment groups and each treatment was performed in two replicates. The exposure period was 30 d and other details were in accordance with those of the acclimation period. Every 3 d, one third of the exposure solution was replaced with new nitrite and MC-LR to maintain concentrations close to the target concentrations. Nitrite concentrations within experimental tanks were measured by using the *N*-(1-Naphthyl) ethylenediamine dihydrochloride spectrophotometric method described in a previous study [76], and MC-LR concentrations were determined using a commercial enzyme linked immunosorbent assay (ELISA) kit (Beacon Analytical Systems, Inc., Saco, ME, USA), from Hou et al. [75].

After the exposure, fish were euthanized with 0.02% (tricaine methanesulfonate) MS-222 (Sigma-Aldrich, St. Louis, MO, USA), and blood samples were collected from the caudal vein to separate the serum. The spleen for each fish was excised and weighed to determine the spleen index, which was calculated following the formula [Spleen index = spleen weight (g)/body weight (g) × 100%]. Serums from 10 individual fish were pooled together as one replicate for the measurement of complement C3 in each treatment group. Six spleens were pooled as one replicate and then stored at −80 °C for the determination of biochemical parameters and gene expression. For pathological analysis, six spleens from fish were fixed for a light and transmission electron microscopic study. All experimental operations were approved by the guidelines of the Institutional Animal and Care and Use Committees (IACUC) of Huazhong Agricultural University (permission number: HZAUFI-2017-008, date of approval: 5 January 2017), Wuhan of China.

### 5.3. Biochemical Parameter Analysis

Spleen samples were homogenized in ice-cold 0.9% NaCl (1:9, *w*/*v*). The extracts were centrifuged for 10 min (3000× *g*, 4 °C) and the supernatant was separated as enzyme source. The analysis of biochemical parameters (MDA, T-AOC and GSH) was performed using commercial kits from the Nanjing Jiancheng Bioengineering Institute, Nanjing, China. Protein concentrations of homogenates were determined by a BCA protein assay [77]. MDA was used as an index of lipid peroxidation, and was determined by thiobarbituric acid reaction method from the study of Ohkawa et al. [78]. The assay of T-AOC was calculated by the absorbance of the 2,2′-azino-bis (3-ethylbenzthiazoline-6-sulfonic acid) radical cation [79]. GSH content was determined using 5,5′-dithiobis-(2-nnitrobenzoic acid) as a substrate to measure the absorbance variation at 412 nm according to the Griffith [80] method. 

Blood samples were centrifuged for 15 min (3000× *g*, 4 °C) and the serum was isolated for immediate use or stored at −80 °C until analysis. Complement C3 content in the serum was measured by immunoturbidimetry, from Jiang et al. [81].

### 5.4. Gene Transcription Analysis

Total RNA was extracted from spleen samples using TRIzol reagent (TaKaRa, Dalian, China). The quality of total RNA was measured by using NanoDrop ND-2000 spectrophotometer (Thermo Scientific, Wilmington, DE, USA). For each sample, 1 μg of total RNA from each sample was reverse transcribed by Takara RT reagent Kit (TaKaRa, Dalian, China). Quantitative real-time PCR (qPCR) was performed on the LightCycler^®^ 480 Real-Time PCR Detection System (Roche, Basel, Switzerland) with SYBR Green Kits (Takara, Dalian, China). Gene specific primer sequences were designed by Primer 5.0 (Appendix A). Amplification conditions were 95 °C for 30 s, followed by 40 cycles of 95 °C for 10 s, 58 °C for 20 s and 72 °C for 10 s. The housekeeping gene *gapdh* was used as an internal control for normalization based on our previous study [75]. The 2^–ΔΔ*C*t^ method was used to calculate the relative expression ratio [82]. All experiments were performed in three biological replicates and two technical replicates.

### 5.5. Pathological Studies

Pathological analysis of the spleen was performed in accordance with our previous studies with little revision [75,83]; more detailed descriptions were provided in the Appendix A.

### 5.6. Statistical Analysis

Statistical analysis was conducted using SPSS 20.0 (SPSS Inc., Chicago, IL, USA) for Windows. All values were expressed as mean ± standard error (SEM). All parameters were evaluated by a two-way (Nitrite and MC-LR) analysis of variance (ANOVA) followed by Duncan’s multiple range test. A Spearman correlation analysis was carried out to determine the relationship between the antioxidant parameters and innate immune parameters. Significant differences were established at *p* < 0.05.

## Figures and Tables

**Figure 1 toxins-10-00512-f001:**
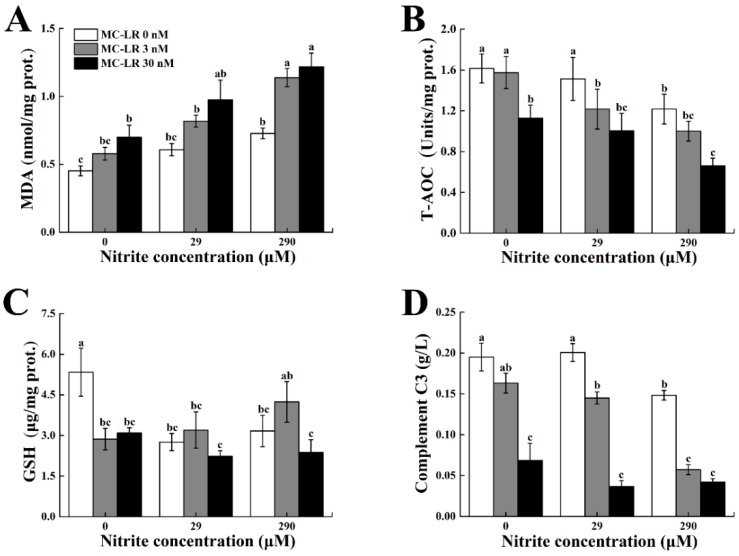
Alterations in splenic malondialdehyde (MDA) (**A**), total antioxidant capacity (T-AOC) (**B**), glutathione (GSH) (**C**) and serum C3 (**D**) of male zebrafish exposed to single and combined concentrations of nitrite and MC-LR for 30 d. Different letters above bars represent significant differences (*p* < 0.05).

**Figure 2 toxins-10-00512-f002:**
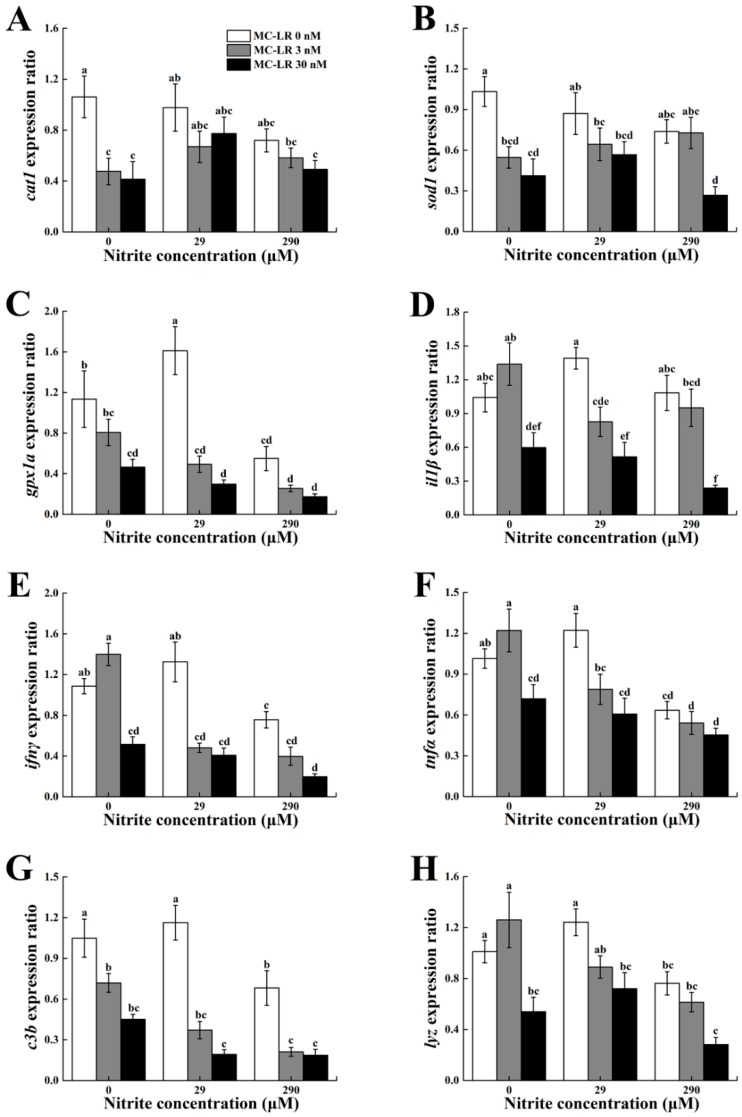
Real-time PCR analysis of mRNA expression levels of *cat1* (**A**), *sod1* (**B**), *gpx1a* (**C**), *il1β* (**D**), *ifnγ* (**E**), *tnfα* (**F**), *c3b* (**G**) and *lyz* (**H**) in the spleen of male zebrafish exposed to single and combined concentrations of nitrite and microcystin-leucine arginine (MC-LR) for 30 d. Different letters above bars represent significant differences (*p* < 0.05).

**Figure 3 toxins-10-00512-f003:**
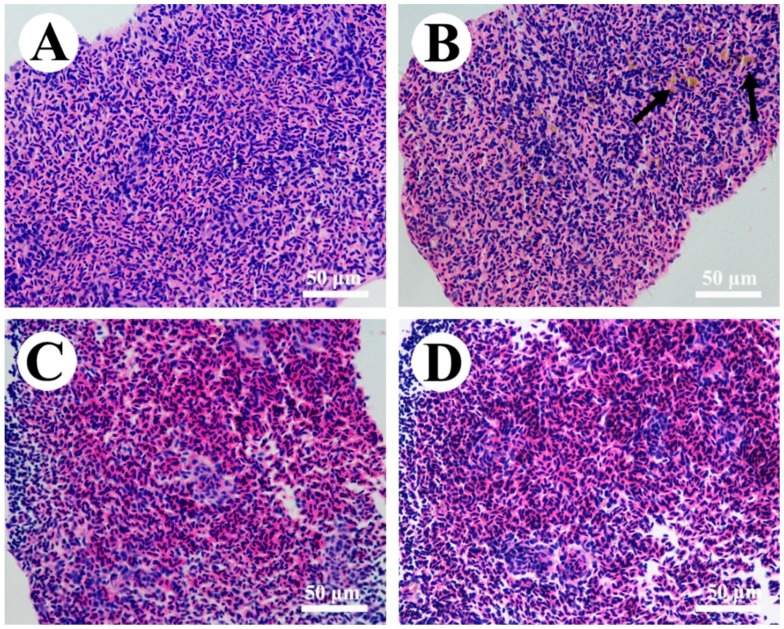
Hematoxylin and eosin (H&E)-stained spleen sections of male zebrafish exposed to single and combined concentration of nitrite and microcystin-leucine arginine (MC-LR) for 30 d. (**A**) Control; (**B**) MC-LR-30 nM; (**C**) Nitrite-290 μM; (**D**) Nitrite-290 μM + MC-LR-30 nM. Black arrow, melano-macrophage centers.

**Figure 4 toxins-10-00512-f004:**
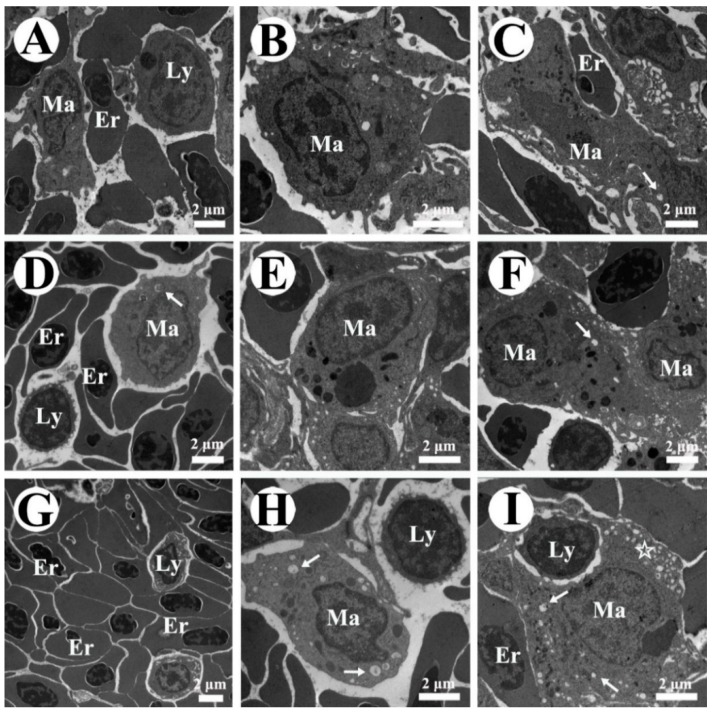
Ultrastructure changes in the spleen of male zebrafish exposed to single and combined concentration of nitrite and microcystin-leucine arginine (MC-LR) for 30 d. (**A**) Control; (**B**) MC-LR-3 nM; (**C**) MC-LR-30 nM; (**D**) Nitrite-29 μM; (**E**) Nitrite-29 μM + MC-LR-3 nM; (**F**) Nitrite-29 μM + MC-LR-30 nM; (**G**) Nitrite-290 μM; (**H**) Nitrite-290 μM + MC-LR-3 nM; (**I**) Nitrite-290 μM + MC-LR-30 nM. White arrow, edematous mitochondria; white star, cell pseudopodia degradation. Other abbreviations: Er, erythrocyte; Ly, lymphocyte; Ma, macrophage.

**Table 1 toxins-10-00512-t001:** Results of two-way ANOVA on the interactions between nitrite and microcystin-leucine arginine (MC-LR) on the spleen index, antioxidant parameters and complement C3 of male zebrafish after 30 d exposure.

Parameters	Source of Variation	*df*	*F*	*p*
Spleen index	Nitrite	2	4.017	0.025
	MC-LR	2	5.145	0.010
	Nitrite × MC-LR	4	0.253	0.907
MDA	Nitrite	2	12.707	<0.001
	MC-LR	2	8.872	<0.001
	Nitrite × MC-LR	4	1.539	0.042
T-AOC	Nitrite	2	6.872	0.002
	MC-LR	2	9.512	<0.001
	Nitrite × MC-LR	4	0.282	0.888
GSH	Nitrite	2	2.679	0.040
	MC-LR	2	3.792	0.030
	Nitrite × MC-LR	4	3.070	0.026
C3	Nitrite	2	90.553	<0.001
	MC-LR	2	15.423	<0.001
	Nitrite × MC-LR	4	4.814	0.005

Abbreviations: MDA, malondialdehyde; T-AOC, total antioxidant capacity; GSH, glutathione; *df*, degrees of freedom; *F*, F-crit; *p*, *p*-value.

**Table 2 toxins-10-00512-t002:** Results of the two-way ANOVA on the interaction between nitrite and microcystin-leucine arginine (MC-LR) on relative mRNA expression levels of antioxidant enzymes and immune-related genes of male zebrafish spleen after 30 d of exposure.

Genes	Source of Variation	*df*	*F*	*p*
*cat1*	Nitrite	2	2.249	0.117
	MC-LR	2	7.856	0.001
	Nitrite × MC-LR	4	1.313	0.280
*sod1*	Nitrite	2	1.035	0.281
	MC-LR	2	12.663	<0.001
	Nitrite × MC-LR	4	1.999	0.111
*gpx1a*	Nitrite	2	11.258	<0.001
	MC-LR	2	24.901	<0.001
	Nitrite × MC-LR	4	3.886	0.009
*il1β*	Nitrite	2	2.331	0.109
	MC-LR	2	24.154	<0.001
	Nitrite × MC-LR	4	2.744	0.040
*ifnγ*	Nitrite	2	19.295	<0.001
	MC-LR	2	39.856	<0.001
	Nitrite × MC-LR	4	11.660	<0.001
*tnfα*	Nitrite	2	15.468	<0.001
	MC-LR	2	6.907	0.002
	Nitrite × MC-LR	4	2.962	0.030
*c3b*	Nitrite	2	12.151	<0.001
	MC-LR	2	58.140	<0.001
	Nitrite × MC-LR	4	2.593	0.049
*lyz*	Nitrite	2	10.311	<0.001
	MC-LR	2	15.744	<0.001
	Nitrite × MC-LR	4	5.762	0.001

Abbreviations: *df*, degrees of freedom; *F*, F-crit; *p*, *p*-value.

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
