# Peer review of "Nitrite Enhances MC-LR-Induced Changes on Splenic Oxidation Resistance and Innate Immunity in Male Zebrafish"

_toxins, 2018, doi:10.3390/toxins10120512_

Reviewer 1 Report

In the results section 2.1., even though there is a SI document, it would be good to give specific numbers and information, rather than a vague statement.

In Fig-1 and Fig-2 can you explain what do the letters and the significant differences mean? They just look like error bars.

It would be good to include some details about the MDA, etc testing. Did the authors follow the cited methods exactly?

Author Response

Response to Reviewer 1 Comments

Please NOTE: the black words are comments or questions by the referees, and the red words are our replies and the blue words are the part that we revised in the manuscript.

Point 1: In the results section 2.1., even though there is a SI document, it would be good to give specific numbers and information, rather than a vague statement.

Response 1: Thanks for your question. According to your suggestion, we added detailed information of these two toxin concentrations in water samples as follow,

        Line 82-83, we changed the sentence of “The results showed nitrite and MC-LR concentrations in the test solutions were stable and relatively close to their nominal concentrations during the whole exposure period” into “The results showed the measured concentrations of nitrite at 29 μM and 290 μM varied from 26.8 μM to 33.3 μM and from 258.1 μM to 332.1 μM, respectively. The real concentrations of MC-LR at 3 nM and 30 nM varied from 2.8 nM to 3.4 nM and from 26.8 nM to 33.5 nM, respectively. The maximum deviation between measured concentrations of nitrite or MC-LR and the corresponding nominal concentrations in each treatment group was less than 20%, which proved the concentrations of these two contaminants within the experimental tanks were relatively stable during the whole exposure period.”

Point 2: In Fig-1 and Fig-2 can you explain what do the letters and the significant differences mean? They just look like error bars.

Response 2: Thanks for your question. In our study, Duncan's post-hoc test after two-way   ANOVA was utilized to analyse differences between treatment groups. Different letters above bars represented significant differences between those treatment groups at the level of P < 0.05, while the same letters above bars indicated no significant difference between those treatment groups.

Point 3: It would be good to include some details about the MDA, etc testing. Did the authors follow the cited methods exactly?

Response 3: Thanks for your question. The assay of MDA level in our study was exactly based on the reaction with thiobarbituric acid according to Ohkawa et al. (1979).

        Line 324-325, as you suggested we revised the sentence of “MDA, as an index of LPO, was determined according to the study of Ohkawa et al. [78]” into “MDA was used as an index of lipid peroxidation, and was determined by thiobarbituric acid reaction method according to the study of Ohkawa et al. [78]”.

Reviewer 2 Report

The study describes interactions between nitrite and microcystin-LR (MC-LR) with respect oxidative stress and immunosuppression in the spleen of the zebrafish.  The study is mostly straightforward, in terms of methods and results, and is relevant and interesting given (as the authors point-out) that animals are likely to be exposed to multiple “toxic” compounds including, as per this study, nitrite and MC-LR.

I had a few questions/comments.

To begin with, are the concentrations of MC-LR and nitrite used in this study at all reflective of levels that would be seen in natural waters? Or are these unrealistically high.  Either way, the environmental relevance of the levels used should be discussed (even if they are not, in fact, what you might expect in natural waters).

In the Results, the authors mention that they measured nitrite and MC-LR, and that they were stable (and approximately equal to nominal concentrations) throughout the experiment.  How was this measured? Although the data are in the Supporting Information, the techniques used should be given in the Methods and Materials.

Similarly, what is “spleen index?” And how is it calculated? This may be common knowledged in certain circles, but the Methods and Materials should describe how it is determined and calculated.

In Figure 1C, it seems that both MC-LR and nitrite may in some cases either increase or decrease GSH with increasing concentration (depending on concentration of the other, i.e., nitrite and MC-LR).  But the authors don’t clearly address this in the text, and in fact, the authors simply and only claim that “GSH exhibited marked decreases” in GSH.

Finally, and perhaps most importantly, the authors keep claiming that interactions between MC-LR and nitrite are “synergistic,” but I am not sure this is true from the data that I see (unless I am missing something).  Synergism to me implies that effects of both treatments combined is great than an additive effect, but looking at effects of most endpoints, I am not sure if the effects are any more that simply additive (if that).  I may be missing some rationale, but the use of “synergistic” needs to be addressed, and if it can’t be qualified, should not be used.

Alongside these comments of the results and other content, there were a few places where the writing and use of English needs to be corrected or improved, although these are mostly minor grammar or style issues (and generally it is concisely and clearly written).  Examples include (although there are others I am sure):

Line 39: “…is considered a classic…”

Line 41: Not sure what is meant by “valued,” and I don’t think this is the correct term.

Line 43:  When the authors say “According to the reports…,” I think they mean according to ONE report, and one particular water body, not in general or always at this level.

Line 51-52: This sentence should be removed.  It is both not really clear, and not a reflective thesis statement of the paragraph.

Line 77: “…coexisting compounds…”

Lines 96-102:  I think this entire paragraph needs to be reorganized a bit, and probably split into two paragraphs (one for T-AOC, and one for GSH).  “On the contrary” (from line 96), for example, should be removed and moved to the sentence in line 99 to read, “In contrast, however, there was no significant interaction…”  And perhaps better to use “decrease” rather than “drop” or “were down,” as the latter two phrases can mean other things, and are ambiguous in this context.

Line 139: “…pseudopodia became degenerative…” perhaps, but not “degeneration.”

Line 170: Should read, “The present study was the first to provide well-founded…”

Line 223: “To date, very few studies…”

Again, there are probably more, and some additional proofreading and editing should be done.

Author Response

Response to Reviewer 2 Comments

Please NOTE: the black words are comments or questions by the referees, and the red words are our replies and the blue words are the part that we revised in the manuscript.

The study describes interactions between nitrite and microcystin-LR (MC-LR) with respect oxidative stress and immunosuppression in the spleen of the zebrafish.  The study is mostly straightforward, in terms of methods and results, and is relevant and interesting given (as the authors point-out) that animals are likely to be exposed to multiple “toxic” compounds including, as per this study, nitrite and MC-LR.

Dear Sir or Madam,

Thank you very much for your rather positive evaluation and useful suggestions on our paper. We hope that our revisions and explanations are proper and satisfactory. The details are as follows,

Point 1: To begin with, are the concentrations of MC-LR and nitrite used in this study at all reflective of levels that would be seen in natural waters? Or are these unrealistically high.  Either way, the environmental relevance of the levels used should be discussed (even if they are not, in fact, what you might expect in natural waters).

 Response 1: Thanks for your question. In fact, environmental concentrations of pollutants include high and low exposure concentrations. In most cases, concentrations of dissolved MCs in water range from 0.1 to 10 μg/L (0.1 to 10 nM) (Singh and Asthana, 2014; Nikitin et al., 2015; Mohamed et al., 2016). In some severely eutrophic lakes like Lake Taihu of China, MCs concentration reached to 35.8 μg/L (36.02 nM) or higher following the collapse of a large, highly toxic blooms (Lahti et al., 1997; Wang et al., 2010), posing an enormous threat to the aquatic animals and human health. In recent studies, a maximum 25 μg/L (25 nM) MC-LR concentration was used as an environmentally relevant concentration (Liu et al., 2014; Wu et al., 2017; Cheng et al., 2017). Similarly, ambient nitrite reached to 20 mg/L (290 μM), for example, nitrite in shore sites of North American Lakes containing decaying algae and plants were in the range of 2-18 mg/L (29-261 μM) (McCoy, 1972; Masser et al., 1999). Based on the above, the levels of MC-LR and nitrite used in the present study are considered to be environmentally related concentrations in cyanobacterial bloom areas.

         In addition, fish spend all their life stages in such aqueous environment, which could well have unsuspected and potential health implications on fish. That is why we are trying to mimic the real natural situation and focuses on the joint toxic effects of MC-LR and nitrite on the immune system of fish in eutrophic water bodies. According to our results in oxidant-antioxidant and innate immune system, single and combined exposure of lower concentrations of nitrite (29 μM) and MC-LR (30 nM) were considered to have adverse health effects on male zebrafish after a 30-d exposure. Then, we added the suggestion into the conclusion section as follow,

        “The present study revealed that long-term exposure of nitrite and MC-LR (greater than or equal to 29 μM and 30 nM, respectively) could cause pathological damages, induce oxidative stress and inhibit innate immunity in the spleen of male zebrafish, indicating that fish might encounter severe immunotoxic effects during the degradation of cyanobacterial blooms.”

Point 2: In the Results, the authors mention that they measured nitrite and MC-LR, and that they were stable (and approximately equal to nominal concentrations) throughout the experiment.  How was this measured? Although the data are in the Supporting Information, the techniques used should be given in the Methods and Materials.

Response 2: Thanks for your question. In the section of Materials and Methods Line (308-309), we have described the methods for the determination of nitrite and MC-LR, “MC-LR concentrations within the experimental waters were determined using the MC-LR ELISA kit (Beacon, USA), and nitrite concentrations were detected by spectrophotometric method [76]”. In order to provide a clear description to the reader, we changed the sentence into “Nitrite concentrations within the experimental waters were measured by using N-(1-Naphthyl) ethylenediamine dihydrochloride spectrophotometric method described in previous study [76], and MC-LR concentrations were determined by using a commercial enzyme linked immunosorbent assay (ELISA) kit (Beacon, USA) according to Hou et al. [75]”.

Point 3: Similarly, what is “spleen index?” And how is it calculated? This may be common knowledged in certain circles, but the Methods and Materials should describe how it is determined and calculated.

Response 3: Thanks for your question. Line 311-312, we have described this information in the section of and Materials and Methods. The spleen for each sampling fish was weighed and the spleen index were calculated following the formula [spleen weight (g)/body weight (g) × 100%] in our study.

Point 4: In Figure 1C, it seems that both MC-LR and nitrite may in some cases either increase or decrease GSH with increasing concentration (depending on concentration of the other, i.e., nitrite and MC-LR).  But the authors don’t clearly address this in the text, and in fact, the authors simply and only claim that “GSH exhibited marked decreases” in GSH.

Response 4: Thanks for your question. It seems that GSH content showed a fluctuation trend with increasing exposure concentrations of nitrite and MC-LR. However, no significant differences in GSH levels were detected among other exposure groups except control group. Actually, we can only observe marked deceased GSH in zebrafish exposed to nitrite and MC-LR showed marked decreases compared to control group. In order to avoid the misunderstanding, we change this sentence as follow,

        “GSH content showed either increase or decrease trend with increasing exposure concentrations of nitrite and MC-LR. However, no significant differences in GSH content were detected among other exposure groups except control group. Compared with the control, the contents of splenic GSH in zebrafish exposed to nitrite or MC-LR exhibited marked decreases”.

Point 5: Finally, and perhaps most importantly, the authors keep claiming that interactions between MC-LR and nitrite are “synergistic,” but I am not sure this is true from the data that I see (unless I am missing something).  Synergism to me implies that effects of both treatments combined is great than an additive effect, but looking at effects of most endpoints, I am not sure if the effects are any more that simply additive (if that).  I may be missing some rationale, but the use of “synergistic” needs to be addressed, and if it can’t be qualified, should not be used.

Response 5: Thanks for your suggestion. In the present study, two-way analysis of variance (ANOVA) was applied to assess the interaction between Nitrite and MC-LR. From Table 1 and Table 2, our results showed there were statistically significant interactions between nitrite and MC-LR on MDA levels, GSH content and C3 levels as well as immune-related genes expression levels (P < 0.05). In consideration of these significant interactions and your suggestion, we replaced the use of “synergistic” with “jointly” in our manuscript.

Point 6: Alongside these comments of the results and other content, there were a few places where the writing and use of English needs to be corrected or improved, although these are mostly minor grammar or style issues (and generally it is concisely and clearly written).  Examples include (although there are others I am sure):

Response 6: Thanks for your suggestion, we revised and reorganized those sentences so as to provide a clear description to the reader. The detailed revisions are as follows,

Point 7: Line 39: “…is considered a classic…”

Response 7: As you suggested we changed this into “…is considered to be a classic…”

Point 8: Line 41: Not sure what is meant by “valued,” and I don’t think this is the correct term.

Response 8: The “valued” means we should pay more attention to the post-bloom toxicity. According to your suggestion, in Line 40-41, we changed the sentence “In natural waters, post-bloom toxicity resulted by the subsequent physical and bacterial degradation should be highly valued” into “In particular, more attention should be paid to post-bloom toxicity resulted by the subsequent physical and bacterial”.

Point 9: Line 43:  When the authors say “According to the reports…,” I think they mean according to ONE report, and one particular water body, not in general or always at this level.

Response 9: We deleted the phrase “According to the reports” as you suggested.

Point 10: Line 51-52: This sentence should be removed.  It is both not really clear, and not a reflective thesis statement of the paragraph.

Response 10: As you suggested we deleted this sentence.

Point 11: Line 77: “…coexisting compounds…”

Response 11: We corrected as you suggested.

Point 12: Lines 96-102:  I think this entire paragraph needs to be reorganized a bit, and probably split into two paragraphs (one for T-AOC, and one for GSH).  “On the contrary” (from line 96), for example, should be removed and moved to the sentence in line 99 to read, “In contrast, however, there was no significant interaction…”  And perhaps better to use “decrease” rather than “drop” or “were down,” as the latter two phrases can mean other things, and are ambiguous in this context.

Response 12: Thanks for your question, we corrected as you suggested, the detailed revisions could be found in revised manuscript.

Point 13: Line 139: “…pseudopodia became degenerative…” perhaps, but not “degeneration.”

Response 13: We corrected as you suggested.

Point 14: Line 170: Should read, “The present study was the first to provide well-founded…”

Response 14: We corrected as you suggested.

Point 15: Line 223: “To date, very few studies…”

Response 15: We corrected as you suggested.

Point 16: Again, there are probably more, and some additional proofreading and editing should be done.

Response 16: As you suggested, we rechecked our manuscript carefully. The detailed revisions could be found in revised manuscript.
